# Meropenem Model-Informed Precision Dosing in the Treatment of Critically Ill Patients: Can We Use It?

**DOI:** 10.3390/antibiotics12020383

**Published:** 2023-02-13

**Authors:** Letao Li, Sebastiaan D. T. Sassen, Tim M. J. Ewoldt, Alan Abdulla, Nicole G. M. Hunfeld, Anouk E. Muller, Brenda C. M. de Winter, Henrik Endeman, Birgit C. P. Koch

**Affiliations:** 1Department of Hospital Pharmacy, Erasmus University Medical Center, 3015 GD Rotterdam, The Netherlands; 2Center for Antimicrobial Treatment Optimization Rotterdam (CATOR), 3015 GD Rotterdam, The Netherlands; 3Rotterdam Clinical Pharmacometrics Group, 3015 GD Rotterdam, The Netherlands; 4Department of Intensive Care, Erasmus University Medical Center, 3015 GD Rotterdam, The Netherlands; 5Department of Medical Microbiology and Infectious Diseases, Erasmus University Medical Center, 3015 GD Rotterdam, The Netherlands; 6Department of Medical Microbiology, Haaglanden Medical Centre, 2597 AX The Hague, The Netherlands

**Keywords:** meropenem, ICU, population pharmacokinetics, model validation, real-world patients, model-informed precision dosing, TDM, sepsis

## Abstract

The number of pharmacokinetic (PK) models of meropenem is increasing. However, the daily role of these PK models in the clinic remains unclear, especially for critically ill patients. Therefore, we evaluated the published meropenem models on real-world ICU data to assess their suitability for use in clinical practice. All models were built in NONMEM and evaluated using prediction and simulation-based diagnostics for the ability to predict the subsequent meropenem concentrations without plasma concentrations (a priori), and with plasma concentrations (a posteriori), for use in therapeutic drug monitoring (TDM). Eighteen PopPK models were included for evaluation. The a priori fit of the models, without the use of plasma concentrations, was poor, with a prediction error (PE)% of the interquartile range (IQR) exceeding the ±30% threshold. The fit improved when one to three concentrations were used to improve model predictions for TDM purposes. Two models were in the acceptable range with an IQR PE% within ±30%, when two or three concentrations were used. The role of PK models to determine the starting dose of meropenem in this population seems limited. However, certain models might be suitable for TDM-based dose adjustment using two to three plasma concentrations.

## 1. Introduction

Meropenem is a broad-spectrum β-lactam antibiotic with an extensive and strong antibacterial activity, especially for drug-resistant Gram-negative bacteria [1,2]. It is often prescribed to treat patients with severe infections in intensive care units (ICU) [3,4,5]. The pharmacokinetic/pharmacodynamics (PK/PD) index that best predicts in vivo antimicrobial activity is the time during which the unbound plasma concentration remains above the minimum inhibitory concentration (MIC) of the pathogen (*f*T > MIC) [6,7]. A 1-log kill bacterial reduction is associated with a *f*T > MIC of at least 40% for meropenem [8,9]. However, this is still under debate, as some studies have suggested that a higher PK/PD target of 100% *f*T > 1-4x MIC is more appropriate for critically ill patients [10,11]. Currently, the standard dosing regimens are based solely on renal function and might not provide the optimal treatment for ICU patients. The variability in organ function between patients, but also within patients during their ICU stay, is relatively high compared to non-ICU patients. This may affect the PK parameters (e.g., the clearance) of antibiotics that are predominantly excreted renally, such as β-lactam antibiotics. Therefore, a standard meropenem dosing regimen may lead to meropenem concentrations outside of the therapeutic window [12,13].

One method to increase the treatment efficacy is through personalized dosing, for example via model-informed precision dosing (MIPD). The latter uses mathematical models, such as population pharmacokinetic (PopPK) models, to predict drug concentrations based on individual patient characteristics and previous drug concentrations. However, these PopPK models must first be validated in the target population to determine whether they can provide accurate drug concentration predictions. A suitable PopPK model can be used for a priori meropenem concentration prediction, which is based on individual patient characteristics to determine the optimal starting dose, but does not make use of measured drug concentrations [14]. Subsequently, when drug concentrations are available from therapeutic drug monitoring (TDM), this data can be used to further individualize and optimize the PopPK models for more precise predictions and dose adjustments [15].

Meropenem MIPD is gaining attention in clinical practice, and many meropenem PopPK models have been published by different medical centers around the world [16,17,18,19,20,21,22,23,24,25,26,27,28,29,30,31,32,33,34,35,36,37,38,39,40,41]. In addition to adult PopPK models, there are models developed for special populations such as pediatric and neonatal populations, [42,43,44,45,46,47,48,49,50] patients with augmented renal clearance (ARC) and patients with renal replacement therapy (RRT) [51,52,53,54,55,56,57,58]. The population used to develop these models varied widely, both in patient numbers and characteristics. Hence, external evaluation is of the utmost importance before applying these models to new target populations. Although several studies have evaluated pharmacokinetic models of meropenem, the patients included in these studies had a relatively sparse sampling of meropenem concentrations [59,60,61,62].

The aim of the study was to evaluate the applicability of published models to real-world ICU patients in Europe using the MIPD approach.

## 2. Results

### 2.1. Patient Characteristics

A total of 20 patients were included. The patient characteristics are summarized in Table 1. A wide range in white blood cell count (WBC), C-reactive protein (CRP), serum creatinine, and fluid balance were observed between the patients. About one-third of the patients (6/20) received renal replacement therapy (RRT) during meropenem treatment.

### 2.2. Population Pharmacokinetic Model Selection

A total of 18 models were included for evaluation after the literature search and article screening. The summary of these models can be found in Table 2. The meropenem PopPK models included in this study had many differences in model structure (e.g., number of compartments) as well as included covariates. Most meropenem models (13/18) were two-compartment pharmacokinetic models, and most (15/18) included creatinine-related covariates on meropenem clearance—creatinine clearance Cockcroft–Gault (CRCL(CG)), estimated glomerular filtration rate (eGFR), modification of diet in renal disease (MDRD) and serum creatinine (SCR), followed by weight on the volume of distribution—total body weight (TBW) and ideal body weight (IBW) (8/18). The covariates albumin, C-reactive protein (CRP), renal replacement therapy (RRT) and APACHE II score were model specific (Table 2).

### 2.3. Model Evaluations

#### 2.3.1. VPC Results of the Models

Visual predictive checks (VPCs) were used to visually evaluate the fit of the individual models. The VPC results of the eighteen models are shown in Figure 1. The concentration range observed in our clinical data was larger than expected from the model. All models showed an under prediction of the variability. Median peak concentrations (time after infusion 0–1 h) were underestimated in ten models (model 1, 2, 4, 8, 10, 11, 12, 13, 14, 18), where model 12 and 14 performed the worst. In contrast, three models (models 7, 9 and 17) slightly overestimated the median peak concentration. The fit of the median trough concentrations was better than that of the peak concentration, with six models (models 2, 11, 12, 13, 14, 16) underestimating the trough concentrations and one model (model 7) overestimating the trough concentration. For the median concentrations between the peak and trough period (time after infusion 1–6 h), five models (model 3, 7, 9, 17, 18) overestimated the observed concentrations and 1 model (model 12) underestimated the observed concentrations. Overall, models 5, 6 and 15 outperformed the other models based on the VPC fit results for median concentrations. However, these models have relatively wide 95% confidence intervals (shaded area) with respect to the other models.

#### 2.3.2. Goodness-of-fit (GoF) Plots

The population prediction versus observation GoF plots are shown in Figure 2. The population-predicted concentrations versus the observed concentrations were not symmetrically distributed around the unity line (where the prediction is equal to the observation). Overall, models 1 and 3 had relatively good fits, while models 7, 8, 9, 15, 16 and 18 showed relatively poor fits. We found that population-predicted concentrations were more likely to overestimate concentrations below 10 mg/L and underestimate concentrations above 50 mg/L. Of the eighteen models, models 2, 10, 12 and 14 showed a tendency to underestimate the observed concentrations, whereas models 7, 9 and 17 showed a tendency to overestimate concentrations.

The individual prediction versus observation GOF plots are shown in Figure 3. The GoF plots of the individual predictions showed a major improvement compared to the population predictions. Most of the points in Figure 4 were evenly distributed around the unity line. Among these models, models 4, 10 and 13 outperformed the other models and showed a good match between predicted and observed concentrations. Model 4, 10 and 13 were all two-compartment models. Both these three models used CRCL (CG) as a covariate on CL. In addition, model 10 used an APACHE score as covariate on Q and model 13 used total body weight as a covariate on V1.

#### 2.3.3. Bias and Precision of the Population and Individual Predictions

The summary of the population prediction error (PEpred%) for each model is shown in Appendix A. The PE% range of the population prediction between the models was large. Overall, most models seem to underestimate the concentration. The individual prediction errors (PEipred%) are shown in Appendix A, which were similar to the results of Figure 3, where model 4, 10 and 13 performed better than the other models. By combining the performance VPC and GoF plot, we chose five models (model 5, 6, 4, 10, 13) as representative models to compare the population prediction error (PEpred%, with no meropenem plasma samples incorporated into model) and individual prediction error using different number of concentrations (PEipred1–3%, one to three meropenem plasma samples used to optimize the model). The results in Figure 4 showed that with the use of two or more prior concentrations in model 10 and 13, the interquartile range (IQR) of PEipred2–3% were within the acceptable range of ±30%. On the other hand, none of the models had a population PE% IQR within ±30% when no plasma concentrations were available to improve the fit. More detailed information for the individual prediction error can be found in Appendix A.

### 2.4. Simulation of Pharmacokinetic Profiles

To evaluate the differences in the pharmacokinetic profiles of the models, simulations were performed with each model for three standard patients with different renal functions (normal renal function, enhanced renal function, and impaired renal function). The results of the simulations of steady-state concentrations (30–40 h after dose) are shown in Figure 5. In patients with normal renal function (Figure 5a), the simulated steady-state peak meropenem concentrations ranged from 25 to 78 mg/L, while the measured steady-state trough concentrations ranged from 0.07 to 9 mg/L. For the simulation of patients with enhanced renal function (Figure 5b), all but three models (model 5, model 9 and model 16) showed lower concentrations compared to the patients with normal renal function, with steady trough concentration ranges from 0.003 to 7.1 mg/L. For simulations in patients with impaired renal function (Figure 5c), drug concentrations were higher in patients with impaired renal function than in patients with normal renal function, with peak concentrations ranging from 42–127 mg/L and trough concentrations ranging from 2.4–45 mg/L.

## 3. Discussion

This is the first study of meropenem MIPD applied to real-world critically ill adult patients in Europe. According to our results, none of the eighteen models seem to be able to provide relatively accurate initial dose guidance without plasma concentrations using only the model with the patient characteristics as covariates. For TDM-based dose adjustment, only two models (models 10 and 13) showed results within the acceptable range of ±30%, although this required at least two concentrations to be included. This demonstrates the importance of model validation prior to use. To guide meropenem dosing in critically ill patients, prior drug concentrations may be necessary to provide adequate predictions.

The results of the simulation of these models showed that the predictive performance of the pharmacokinetic models of meropenem varied widely. Although most of these models were based on critically ill patients, the model structure and associated covariates were diverse. Protein binding would most likely not be a major factor as only ~2% is bound to plasma proteins [63]. Based on the results of the simulation, the effect of increased renal clearance on meropenem concentrations appeared to be less than that of impaired renal function. The greater emphasis on impaired renal function in these models may be due to the fact that patients in these studies were more likely to have a decreased rather than an increased renal function compared with normal individuals. As shown in Table 2, although the creatinine clearance in the different studies ranged from 3–357 mL/min, the median value of renal clearance in these models ranged from 30 to 100 mL/min, which is lower than normal renal function. Most models implemented a form of creatinine clearance as a covariate on meropenem clearance (e.g., MDRD, CRCL Cockcroft–Gault, CKD-EPI). However, these estimations are not a good representation of the actual creatinine clearance in critically ill ICU patients. More precise estimation methods for renal clearance might further reduce model misspecification [64].

The VPC results and population prediction versus observation GoF plots showed that the models provided a poor fit in the absence of information regarding individual drug concentrations. This suggested that these models could not accurately guide the initial doses in our patients. Although model 1, 5, 6, 8, 11 and 13 have been externally validated in other studies [51,59,60,61], the models were not able to fit our data well. This might be due to possible differences in our patient baseline/characteristics. In our study, our patients appeared to be divided into two groups, one with relatively high concentrations and the other with relatively low concentrations. However, the variability/dispersion of concentrations in our study could not be differentiated by using patients’ renal clearance or any of the other covariates in the published models. This may explain the underperformance of all models and the greater likelihood of overestimating concentrations below 10 mg/L and underestimating concentrations above 50 mg/L in the population. The fit of the models using individual plasma concentrations showed a vast improvement. This suggested that some models may be suitable for TDM-based dose adjustment. However, only a small number of models showed good predictive performance. It is therefore of utmost importance to validate the model on the intended target population.

The results of the predictive power accuracy analysis of the five most representative models were consistent with those of the VPC and GoF plots. None of these models had good a priori predictive power to discriminate between potentially high- or low-concentration groups in our ICU population. Therefore, we should be cautious about using the current PopPK models to guide individualized meropenem dosing in ICU patients during the initial dosing phase in the absence of meropenem concentrations. Without samples, the range of the prediction error was very large, ranging from −80% to 500%. However, when combined with TDM, the range of PE% decreased severely. Two models (model 10 and 13) performed the best in our study. These two models showed prediction errors within an acceptable range (PE% IQR within ±30%) when at least two concentrations were available. This might indicate that dense sampling (at least two samples per dosing occasion) for meropenem TDM in critically ill patients might be necessary to have adequate predictions and therefore more reliable dose-adjustment recommendations.

Our results were similar to those of Yang N. et al. [62], who showed that the current models were not suitable for daily meropenem dosing guidance for ICU patients. However, there were still some differences. First, the characteristics of ICU patients in hospitals in the Netherlands may differ from those in Asian hospitals. This might explain the differences in PE% range for models which were evaluated in both studies (model 1, 5, 6, 11, 13). In addition, the blood samples in our dataset were sampled more densely, which provided additional areas of the pharmacokinetic profile of meropenem. In the model prediction analysis, up to three plasma concentrations were available to improve the model fit and compare model performance. Lastly, our study included additional models for evaluation, and since we used NONMEM software for coding, we only included parametric models to ensure reliable results. The included models were all developed in a population in the ICU to reduce the amount of extrapolation to our population of critically ill ICU patients.

In summary, the models that were available from the literature performed poorly in our population, especially in the absence of individual meropenem concentrations, hence implying that MIPD is not suitable for guiding patients’ doses based purely on model and patient characteristics. Hence, it will most likely not perform well at initiation of the treatment when no meropenem concentrations are available. The number of patients were limited; however, if the fit is poor for these real-world patients, it is unlikely that the performance of the total population would be adequate. However, two of these models provided predictions in the acceptable range of ±30% when at least two or more concentrations were used to tailor the model fit. The use for TDM purposes, by taking meropenem concentrations into account, might be suitable. However, this warrants further confirmation, for example via a prospective study. The study also confirms the importance of TDM and model validation for the implementation of meropenem PopPK models in ICU patients.

Our study had some limitations. First, the sample size was limited and data was collected on one dosing occasion. Therefore, the effect of variability between occasions could not be taken into account. Second, since this study was limited to parametric models, future work can be extended by nonparametric models.

## 4. Materials and Methods

### 4.1. Clinical Data

A total of 86 blood samples from 20 ICU patients, obtained in the EXPAT trial (Netherlands Trial Registry, NTR 5632), was used for model external validation [13]. The study protocol was approved by the Erasmus MC Medical Ethics Committee (EXPAT, NL53551.078.15) and informed consent was obtained from all patients or their legal representatives [13]. All enrolled patients were admitted to the ICU between January 2016 and June 2017 and received intermittent intravenous meropenem (2–4 g per day) for the treatment of an infection. On day 2 after the start of meropenem treatment, five venous blood samples were collected: 15–30 min before the next dose infusion (trough concentration, C_min_), 15–30 min after the end of infusion (peak concentration, C_max_), 1 h and 3 h after the end of infusion, and at 15–30 min before the administration of the next dose (second C_min_). In addition to the samples used for pharmacokinetic analysis, the following additional data were collected: patient demographics (age, sex, and actual/ideal body weight), patients’ diagnosis, renal replacement therapy (type of RRT and RRT session timetables), fluid balance (input/output), APACHE II and SOFA score, and biochemistry results. Creatinine clearance was calculated using the Cockcroft–Gault (CG) equation [65], glomerular filtration rate with MDRD [66], and CKD-EPI formula [67]. All this information was recorded at baseline (prior to sampling) and on the day of sampling.

Meropenem concentrations in blood samples were quantified as previously described [68]. All samples were thawed 30 min before analysis and analyzed using an ultra-performance liquid chromatography-tandem mass spectrometry (UPLC-MS/MS) method. A total of 50 μL of plasma was mixed with 150 μL of formic acid and 800 μL of internal standard solution. After samples were centrifuged and transferred to auto-sampler vial, 4 μL of extract was injected into the UPLC-MS/MS system. The lower limit of quantification (LLOQ) and upper limit of quantification (ULOQ) were 0.174 mg/L and 34.8 mg/L, respectively.

### 4.2. Population Pharmacokinetic Model Selection

A literature search was conducted (up to June 2022) on PubMed and Embase database to assess the available literature on meropenem PopPK models in adult ICU patients. The models in the publications were manually screened. The inclusion criteria were: parametric population pharmacokinetic models, adults and ICU patients, PK model code or sufficient information to rebuild the model in NONMEM. The exclusion criteria were: nonparametric models, pediatric or neonatal patient population models, burn patient population models, steady-state models missing volume of distribution parameter and some other models that use covariates that are not available in our population. Detailed search strategy can be found in Appendix A. All the selected meropenem models were encoded and analyzed using NONMEM^®^ 7.4 (Icon Development Solutions, Hanover, MD, USA).

### 4.3. Model Evaluations

#### 4.3.1. VPC Results of the Models

The global fit of the population pharmacokinetic models were evaluated using visual predictive checks (VPCs) generated in Perl speaks NONMEM (PsN) (version 4.7.0) [69], and the data were later analyzed using the Xpose4 R package (version 1.0.1) in R (version 4.0.5). VPCs were further analyzed by stratifying patients on whether or not they received renal replacement therapy (RRT). For each model, 500 iterations of simulations were used.

#### 4.3.2. Goodness-of-Fit Plot

The diagnostic goodness-of-fit (GOF) plots were implemented for model comparison, plotted using R package Xpose4. The population-predicted concentration versus observed concentration and the individual-predicted concentration versus observed concentration were plotted for each model.

#### 4.3.3. Bias and Precision of Model Predictions

Bias was assessed using relative prediction error (PE%), determined by the difference between observed concentration (Cobs) and population-predicted concentration (Cpred) or individual-predicted concentration (Cipred). The population prediction error (PEpred%) and the individual prediction error (PEipred%) were calculated using Equation (1) and Equation (2), respectively [70,71].
(1)$$\mathrm{PEpred}\%=\frac{\mathrm{Cpred}-\mathrm{Cobs}}{\mathrm{Cobs}}\times 100\left(\%\right)$$
(2)$$\mathrm{Peipred}\%=\frac{\mathrm{Cipred}-\mathrm{Cobs}}{\mathrm{Cobs}}\times 100\left(\%\right)$$

The PE% was used for evaluating the prediction accuracy of each model. The Pepred% indicated the prediction accuracy after initial dosing with no prior concentration information, based solely on patient characteristics. The PEipred_1–3_% implied a prediction accuracy of TDM-based dose adjustments by taking one to three concentrations into account. The PE% was considered acceptable if the IQR of PE% was within 30%.

The root mean square error (RMSE) was used to evaluate the model precision. The RMSE was computed with the PE% to show the how far the predicted values were from the observed values in a regression analysis (Equation (3)) [72].
(3)$$\mathrm{RMSE}=\sqrt{(\sum \mathrm{PE}{\%}^{2})/n}$$

### 4.4. Simulation of Pharmacokinetic Profiles

For each selected/screened population pharmacokinetic model, the pharmacokinetics of meropenem were simulated under three typical renal function statuses (normal renal function creatinine clearance (CRCL) of 100 mL/min, augmented renal function CRCL of 200 mL/min, and impaired renal function CRCL of 20 mL/min) in standard patients. Apart from the renal function difference, the standard patient’s conditions were the same for all three patients: male, 55 years of age, weight of 80 kg, ideal body weight of 70 kg, height of 180 cm, albumin level of 25 g/L, APACHE score of 24, with sepsis. The simulated administration method of meropenem was 3 times a day, 1 g each time, intravenously infused with an infusion duration of 30 min. For models using CRCL as covariate, we used a CRCL(CG) (Cockcroft–Gault equation) estimation.

## 5. Conclusions

The predictive performance of the evaluated PK models available from the literature showed high variability and poor predictive power in our population of critically ill patients in the absence of individual meropenem concentrations. This might be due to the heterogeneity of this population with models that are probably too stringent. However, the use of individual meropenem concentrations showed a major improvement. Two models provided predictions within an acceptable range of ±30%, although this required at least two plasma concentrations to improve the model fit. In this population, the PK models did not seem well suited to predict the starting dose (with no prior plasma concentrations), but might be useful for TDM purposes when at least two samples are available. However, this is based on a limited number of patients from a very heterogenetic population. This study also shows the importance of validating models on the intended target population, since the performance of different models can vary to a large extent.

## Figures and Tables

**Figure 1 antibiotics-12-00383-f001:**
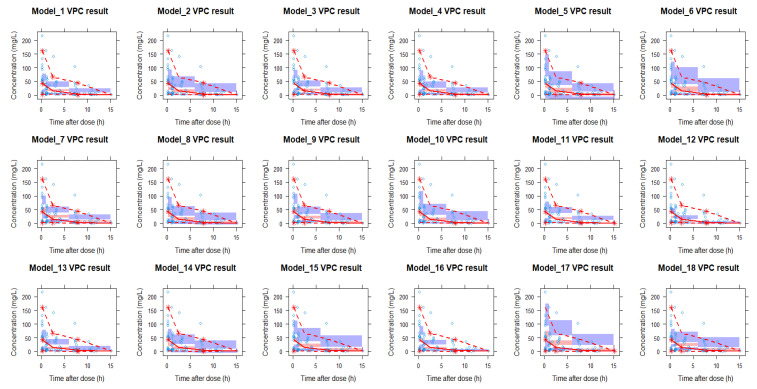
Observed meropenem concentration–time data and the visual predictive check (VPC) of the 18 models. The blue brackets are the observed concentrations. The red line is the observed median and the two blue lines are the 5th and 95th percentiles of the observed data. The red shaded area is the 95% CI of the model predicted median and the purple shaded areas are the 95% CIs of the model-predicted 5th and 95th percentiles. Red lines above the shaded areas represents model underprediction and red lines under the respective areas represents overprediction of the models at those intervals Additionally, VPCs were stratified (Appendix A) on whether patients received renal replacement therapy or not (RRT versus non-RRT). Overall, the observed concentrations in the non-RRT group were lower than in the RRT group. In the non-RRT group, the median peak concentrations (at time after infusion 0–1 h) were underestimated for six models (model 2, 8, 10, 11, 12, 14, 18) and overestimated for three models (models 7, 9 and 17). Median trough concentrations (time after 6–15 h post-infusion) showed a better fit, with models 11 and 18 underestimating concentrations and model 7 overestimating concentrations. For the median concentrations between the peak and trough (time after infusion 1–6 h), thirteen models (model 1, 3, 4, 6, 7, 8, 9, 11, 13, 15, 16, 17, 18) overestimated the observed concentrations while one model (model 12) underestimated the observed concentrations. For the VPC plot of the RRT patients, the median peak concentrations (time after infusion 0–1 h) were underestimated in eight models (model 1, 2, 4, 10, 11, 12, 13, 18). For the median intermediate time concentrations (time after infusion 1–6 h) and median trough concentrations (time after infusion 6–15 h), all but model 14 and 17 underestimated the observed concentrations. Compared with the combined VPC analysis, model 5 performed best in the non-RRT group while model 14 performed best in the RRT group.

**Figure 2 antibiotics-12-00383-f002:**
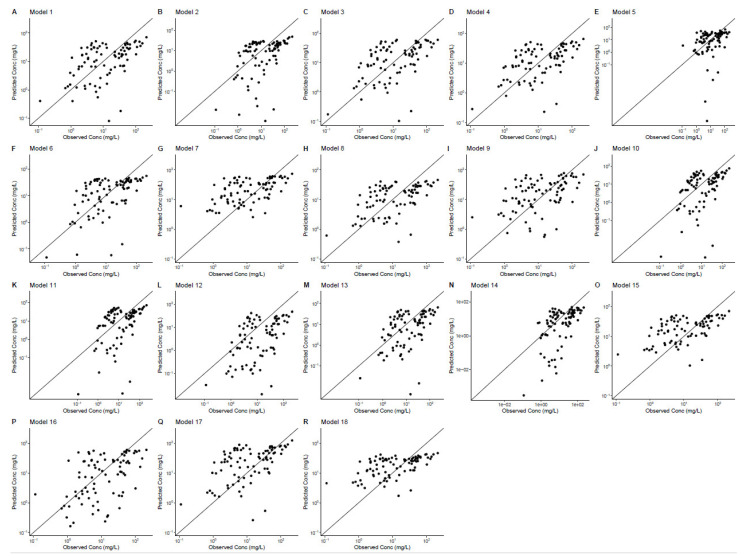
The goodness-of-fit plots of the 18 models. Population predictive concentration (y-axis) versus observed concentration (x-axis).

**Figure 3 antibiotics-12-00383-f003:**
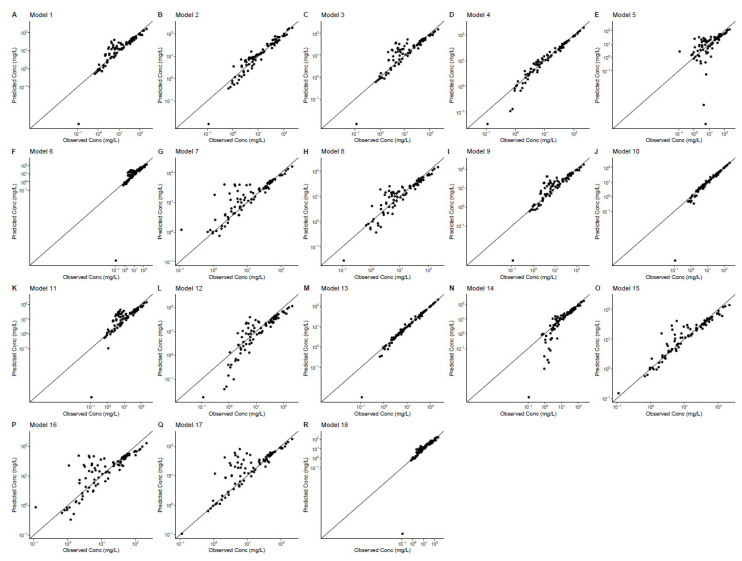
Individual predictive goodness-of-fit plots for eighteen models. Individual-predicted concentrations (y-axis) versus observed concentrations.

**Figure 4 antibiotics-12-00383-f004:**
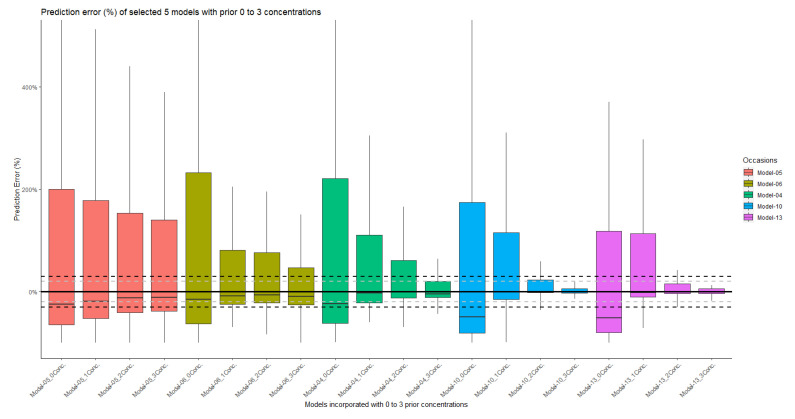
Prediction errors of model 5, 6, 4, 10 and 13. For each model, none, one, two or three plasma concentrations were used to improve the model fit. The grey dashed line is the y = ±20%. The black dashed line is the y = ±30%.

**Figure 5 antibiotics-12-00383-f005:**
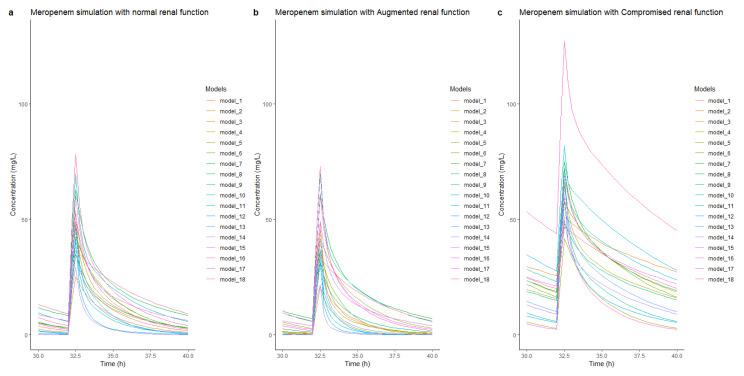
Steady-state concentrations and time simulation of the three typical patients with different renal functions: normal renal function (**a**), augmented renal function (**b**), and impaired renal function (**c**). The simulations were all based on a standard patient (55 years, 180 cm, 80 kg, APACHE II score 24, albumin 25 g/L, serum creatinine 100 umol/L). All received a meropenem dose of 1 g q8 h for 40 h. Steady-state concentrations were shown at the fourth dose (30–40 h after first dose).

**Table 1 antibiotics-12-00383-t001:** Patient baseline characteristics.

**Demographic data (median, (range))**	
∘ Age (years)	64.5 (56–70)
∘ Sex (male/female)	16/4
∘ Length (cm)	175 (155–195)
∘ Weight (kg)	80 (40.4–100)
∘ BMI	26.3 (16.8–34)
**Clinical data at inclusion (median, (range))**	
∘ SOFA	7.5 (3–16)
∘ APACHE II	22.5 (9–33)
∘ Albumin (g/L)	25 (12–38)
∘ Serum creatinine (μmol/L)	108 (33–464)
∘ Temperature (°C)	36.7 (36.1–39.3)
∘ WBC (×10^9^/L)	13.7 (0.05–84)
∘ CRP (mg/L)	194.5 (16–495)
∘ Serum urea (mmol/L)	12.7 (3.3–45.6)
∘ eGFR (mL/min/1.73 m^2^)	53 (11–149)
Fluid balance (mL)	735 (−3142–7566)
**Clinical outcomes (median, (range))**	
∘ ICU LOS (days)	10.5 (2–55)
∘ 30-day mortality	4 (20%) *
RRT	6 (30%) *

Abbreviations: BMI: Body Mass Index; SOFA: sequential organ failure assessment; APACHE II, Acute Physiology and Chronic Health Evaluation version 2; WBC: white blood cell; CRP: C-reactive protein; eGFR: estimated glomerular filtration rate (CKD-EPI); ICU: intensive care unit; LOS: Length of stay; RRT: renal replacement therapy. Asterisk *: number (percentage).

**Table 2 antibiotics-12-00383-t002:** Characteristics of included meropenem population pharmacokinetic models.

Reference	Year	Patient Population	RRT	Nr. of Patients	Nr. of Samples	Covariates	Typical CL and Q Value (L/h)	Typical V Values (L)	Renal Function (mL/min)	Weight (kg)
1.Lisa Ehmann et al.	2019	Critically ill patients with severe infections	Non-RRT	48	1376	CL: CLCR(CG); Vc: TBW, Albumin	CL: 9.25 Q:28.4	Vc:7.89 Vp:16.1	80.8 (CI 95%: 39.4–170)	70.5 (CI 95%: 47–121)
2. Yong Kyun Kim et al.	2018	Adult patients with sepsis and severe sepsis	Non-RRT	37	148	CL: SCR; Vc: TBW	CL: 16.7	V: 30.7	64.3 (IQR 51.1–97.5)	62.8 (IQR 52.2–69.7)
3. Muhammad Usman et al.	2016	Elderly patients with critical illness in medical and surgical ICU	Contain RRT	178	493	CL: CLCR(CG), IBW; Vc: IBW	CL: 5.27 Q: 9.92	Vc:17.2 Vp:10.6	39.0 (3–231.4)	75 (37–147)
4. Eun Kyoung Chung et al.	2016	Obesity and Non-obesity ICU and hospitalized patients	NA	40	360	CL: CLCR(CG);	CL: 9.13 Q: 15.9	Vc:14.3 Vp:17.7	82 (SD ± 40)	129 (SD ± 61)
5. Francesca Mattioli et al.	2016	ICU patients with nosocomial infection	Non-RRT	27	118	CL: Sepsis level; V: Albumin, Age	CL: 2.18	V: 8.3	82.9 (43.2–131.6)	68 (SD 76.2 ± 30.3)
6. SutepJaruratanasirikul et al.	2015	ICU patients with severe sepsis or septic shock	NA	9	171	CL: CLCR(MDRD);	CL: 7.82	V: 23.7	59.43 (12.37–214.55)	62.88 (SD ± 11.64)
7. Yoko Niibe et al.	2020	Critically ill with continuous hemodiafiltration	RRT	21	350	CL: eGFR (CKD-EPI)	CL: 4.42 Q: 7.84	Vc: 14.82 Vp: 11.75	26.1 (5–74.8)	70.6 (44–122)
8. Isabelle K. Delattre et al.	2012	ICU patients with severe sepsis or septic shock	Non-RRT	88	418	CL: CLCR(CG), TBW; Vc: TBW,	CL: 9.87 Q: 4.97	Vc: 24.4 Vp:7.01	NA	70 (38–125)
9. Dagan O Lonsdale et al.	2020	Critically ill patients from neonatal to elderly	Contain RRT	47	Na	CL: Age, TBW; V: TBW	CL: 8.7 Q: 13.8	Vc: 8.8 Vp:10.6	* 93.0 (IQR 63.0–134.0)	* 70 (IQR 64.0–92.0)
10 Qing-tao Zhou, Bei He et al.	2012	Elderly patients with lower respiratory tract infection	NA	75	284	CL: CLCR(CG); Q: APACHE	CL: 8.98 Q: 15.9	Vc:16.1 Vp:12	53.4 (8.6–129)	64.4 (SD ± 12.3)
11. Jason A. Roberts et al.	2009	Critically ill with known or suspected sepsis	Non-RRT	10	222 plasma; 274 microdialysis	CL: CLCR(CG)	CL: 13.6 Q: 56.3	Vc:7.9 Vp:14.8	100 (IQR 69–161)	78 (IQR 75–85)
12. Frédéric Frippiat et al.	2015	ICU patients with nosocomial pneumonia	Contain RRT	55	Na	CL: CLCR(CG); V1:TBW	CL: 10.2 Qp: 6.9 Qe: 66.5	Vc:5.2 Vp:12.1 Ve:11.3	NA	78.4 (SD ± 18.4)
13. Chonghua Li et al.	2006	Adult patients with intra-abdominal infections and pneumonia	NA	79	341	CL: CLCR(CG); V1:TBW	CL: 14.6 Q: 18.6	Vc:10.8 Vp:12.6	** SCR (mg/dl):1.0 (0.4–6.9)	70 (40.6–127)
14. Kiran Shekar et al.	2014	ICU ECMO patients with known or suspected sepsis	Contain RRT	21	249	CL: RRT, CLCR(CG)	CL: 5.1 Q: 21	Vc:18.7 Vp:13.2	106 ** (IQR 98–127)	80 ** (IQR 75–85)
15. Jinhua Lan et al.	2022	Critically ill patients with pneumonia	Contain RRT	48	236	CL: eGFR (CKD-EPI)	CL: 7.48 Q: 26	Vc:15.9 Vp:35.3	35.69 (22.30–90.84)	na
16. Yoko Niibe et al.	2022	Critically ill patients with respiratory or intra-abdominal infection	Non-RRT	12	237	CL: CRP	CL: 9.3 Q: 9.7	Vc:12.6 Vp:7.8	95.2 (53.2–357.3)	60.6 (46.0–80.5)
17. Dong-Hwan Lee et al.	2021	ICU ECMO patients with sepsis and nosocomial infections	Contain RRT	26	169	CL: eGFR (CKD-EPI)	CL: 6.37 Q: 10.7	Vc:9.07 Vp:7.91	91.6 *** (IQR 45.6–103)	54.4 *** (IQR 50.5–64.5)
18. Abdullah Alsultan et al.	2021	ICU patients with suspected or proven bacterial infection	Contain RRT	43	86	CL: CLCR(CG); V1:TBW	CL: 6.4	V: 30	95 (IQR 48.5–216)	74 (IQR 50.5–84)

CL: clearance; Vc: central compartment; Vp: peripheral compartment; Ve: epithelial lining fluid compartment; Q: intercompartmental clearance; RRT: renal replacement therapy; CAP: community-acquired pneumonia; HAP: hospital-acquired pneumonia; IAI: intra-abdominal infection; APACHE II, Acute Physiology and Chronic Health Evaluation version 2; TBW: total bodyweight; IBW: ideal body weight; CRP: C-Reactive protein; CRCL(CG): Creatinine clearance (Cockcroft–Gault equation); eGFR (CKD-EPI): CKD-EPI Creatinine equation for estimated glomerular filtration rate; CLCR(MDRD): Creatinine clearance (The original modification of diet in renal disease (MDRD) equation). Usually, the renal function shows the relevant covariates in CL. * This study does not showed the renal function of all the β-Lactam antimicrobial patients. ** no CRCL information, so used only SCR information. *** there are ECMO and non ECMO patients, we only showed patients without ECMO. NA: not available.

## Data Availability

The data that support the findings of this study are available from the corresponding author upon reasonable request.

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
