# Peer review of "Meropenem Model-Informed Precision Dosing in the Treatment of Critically Ill Patients: Can We Use It?"

_antibiotics, 2023, doi:10.3390/antibiotics12020383_

Round 1

Reviewer 1 Report

The article is clear and underlines the inaccuracy of available models in predicting meropenem concentration in ICU. It also showed that the models accuracy improved if fitted after previous TDM . The message is important.

Concerns :

-        The messages about the differences between the accuracy of the available models should be tone down because probably very different from one sample to another. A unique sample of 20 patients could not be considered as representative

-        some points have not been taken into account or discussed

o   PK : CKDEPI or Cockcroft are clearly not representative of creatinine clearance in ICU patients. Glomerular hyperfiltration and variability of creatinine clearance with time in septic patients should be discussed to explain the considerable variability of the observed concentrations

o   PK : free drug vs total drug should also be discussed.

o   PD : for a clinician the treatment should be immediately effective (before the steady state) and should avoid overdose and possible toxicity when patient is stabilized ? how these models answered to these questions ? 

Reviewer 2 Report

The manuscript entitled as “Meropenem model-informed precision dosing in the treatment of critically ill patients: can we use it?” is an interesting and well-structured paper that will be of interest to the readers of this journal. This is a generally well written paper containing a significant volume of research papers. Authors have tried to develop the relation between predicted and estimated dosage regimens of critically ill patients, using the MIPD approach. Manuscript has been well written and explained elaborately in the review. The paper is clear and easy to follow and the value of the individualizing drug therapy is well demonstrated. I just have a single concern mentioned below:

Line no 78-80 “There are several studies which evaluated pharmacokinetic models of meropenem, these studies used limited per patient data.” The authors want to explain something which is not clear by this line. It should be rephrased.

However, the discussion presented in this paper is significant enough to warrant publication in Antibiotics and fully meet the requirements of urgency and novelty, justifying publication as a communication.
